# Transition Metal Dichalcogenides for the Application of Pollution Reduction: A Review

**DOI:** 10.3390/nano10061012

**Published:** 2020-05-26

**Authors:** Xixia Zhang, Sin Yong Teng, Adrian Chun Minh Loy, Bing Shen How, Wei Dong Leong, Xutang Tao

**Affiliations:** 1State Key Laboratory of Crystal Materials, Shandong University, Jinan 250100, China; txt@sdu.edu.cn; 2Central European Institute of Technology, Brno University of Technology, Purkynova 656/123, 612 00 Brno, Czech Republic; 3Institute of Process Engineering & NETME Centre, Brno University of Technology, Technicka 2896/2, 616 69 Brno, Czech Republic; Sin.Yong.Teng@vut.cz; 4Department of Chemical Engineering, Monash University, Clayton, Melbourne 3800, Australia; adrian.loy@monash.edu; 5Research Centre for Sustainable Technologies, Faculty of Engineering, Computing and Science, Swinburne University of Technology, Jalan Simpang Tiga, Kuching 93350, Malaysia; bshow@swinburne.edu.my; 6Department of Chemical and Environmental Engineering, University of Nottingham, Semenyih 43500, Malaysia; ebxwl1@nottingham.edu.my

**Keywords:** transition metal dichalcogenide (TMDCs) nanomaterials, layered materials, nanocatalysis, gas cleaning, catalysis, pollution reduction, emission control

## Abstract

The material characteristics and properties of transition metal dichalcogenide (TMDCs) have gained research interest in various fields, such as electronics, catalytic, and energy storage. In particular, many researchers have been focusing on the applications of TMDCs in dealing with environmental pollution. TMDCs provide a unique opportunity to develop higher-value applications related to environmental matters. This work highlights the applications of TMDCs contributing to pollution reduction in (i) gas sensing technology, (ii) gas adsorption and removal, (iii) wastewater treatment, (iv) fuel cleaning, and (v) carbon dioxide valorization and conversion. Overall, the applications of TMDCs have successfully demonstrated the advantages of contributing to environmental conversation due to their special properties. The challenges and bottlenecks of implementing TMDCs in the actual industry are also highlighted. More efforts need to be devoted to overcoming the hurdles to maximize the potential of TMDCs implementation in the industry.

## 1. Introduction

Transition metal dichalcogenides (TMDCs) are a large family of two-dimensional (2D) layered materials, which are scientifically interesting and industrially important. These materials have attracted tremendous attention because of the unique structural features and interesting properties, such as optoelectronics, electronics, mechanical, optical, catalytical, energy-storage, thermal, and superconductivity properties [1,2,3,4,5,6,7]. TMDCs are the compounds of the chemical formula MX_2_, where M is a transition metal element of groups IV-VII B (Mo, W, V, Nb, Ta, Ti, Zr, Hf, Tc, and Re) and X is a chalcogen element (S, Se, and Te). The X-M-X unit layer consists of three atomic layers, in which one centre atom layer (M) is sandwiched between two chalcogen atom layers (X). TMDCs occupy the layered structures, which resemble that of graphite. The interlayers are stacked by weak van der Waals force, leading to the formation of monolayers or nanolayers from the bulk materials via exfoliation [8]. Different stacking of the layers along c-axis determines polymorphic crystal structures in TMDCs, and the common phases are 1T, 2H, 3R, and Td phases (T—trigonal, H—hexagonal, R—rhombohedral, and Td—distorted octahedral) [9].

There are more than 40 different TMDC types, including metals (such as TiS_2_ and VSe_2_), superconductors (such as TaS_2_ and NbS_2_), semimetals (such as MoTe_2_ and WTe_2_), and semiconductors (MoS_2_, MoSe_2_, WS_2_, and WSe_2_). TMDCs exhibit interesting band structures with tunable bandgaps. The bandgap is one of the most important factors in 2D materials for determining the properties and applications. For instance, graphene is a semimetal with zero bandgap, which limits its applications in electronics and photo-electronics. TMDCs exhibit variable bandgaps from 0 to 3 eV, which can be tuned by thickness [10], defects [11], dopants [12], and mechanical deformations (by applying the tensile strain or compressive strain) [13,14]. The most studied semiconducting TMDCs (e.g., MoS_2_, MoSe_2_, WS_2_, and WSe_2_) have shown typical features in electronic structures. The bandgap increases with the decreasing thickness and it possesses the transition from indirect in the bulk crystals to direct in the monolayers [10,15]. For instance, the indirect bandgap of −1.29 eV will be changed to a direct bandgap of −1.8 eV when bulk MoS_2_ is down to a monolayer [16].

Benefiting from their unique crystal structures and electronic structures, TMDCs have shown great potential in various fields, including electronics/optoelectronics [1,17], catalysis [18], energy storage [19] and conversion [20], sensing [21,22], and so on. The application of TMDCs in pollution reduction is a compelling research topic. The increasing environmental pollution issue has been one of the most serious problems on Earth. Enormous efforts have been made to search the efficient and low-cost methods for addressing the environmental pollution issue. TMDCs may be a kind of promising materials for tackling these problems with several advantages. Firstly, TMDCs have a high surface-to-volume ratio. They offer more effective active sites on the surface, as well as abundant unsaturated surface sites. Thus, the layered TMDCs are excellent platforms for the anchor of semiconductor nanoparticles in various photocatalytic applications [23,24]. Due to their high surface-to-volume ratio, TMDCs are extremely sensitive to the surrounding atmosphere and can be utilized in toxic gas sensing and adsorption. Secondly, TMDCs have tunable bandgaps, which enhances the photocatalytic performance in nanocomposite by offering appropriate bandgap and band alignment [25]. Thirdly, defect engineering can be easy to implement in 2D materials, which have been confirmed to be an efficient method for intensifying the catalytic activities in TMDCs [26,27,28,29]. Lastly but importantly, there is a large variety for TMDCs (about 40 kinds) and they have an abundant amount in nature or can be synthesized [9]. So far, MoS_2_, WS_2_, MoSe_2_, and ReS_2_ have been naturally found [30,31,32]. Specifically, MoS_2_ exists as molybdenite in nature and is the main source of molybdenum with a large amount [33]. The main metals (W and Mo) in TMDCs are both abundant, cheap, and widely used in industry [34]. TMDCs can be prepared by using various techniques, such as chemical vapor deposition (CVD) [35,36], chemical vapor transport (CVT) [37,38], flux growth method [39], hydrothermal synthesis [40], Langmuir−Schaefer deposition [41], etc. In addition, the top-down exfoliated method can be also used to fabricate few-layer TMDCs from bulk crystals, e.g., mechanical exfoliations and liquid phase exfoliations [42,43,44]. With increasing interests in TMDCs for applications, we aim to prepare an overview of the recent progress of TMDCs in reducing the environmental pollution. We will summarize the representative efforts, including gas adsorption and removal, gas sensing, wastewater treatment, fuel cleaning, CO_2_ valorization, and conversion.

## 2. Gas Adsorption and Removal

In recent years, TMDCs have been used as gas removal (via adsorption) in pollution reduction. Figure 1 presents the number of research articles that involved the use of TMDCs in gas adsorption applications. The increasing trend (the total number of related articles published in the year 2019 is more than three-fold than that of the year 2015) indicates the potential of TMDCs in gas removal processes. Based on the bibliology search, MoS_2_ is still the most widely studied TMDCs among the TMDCs considered in this review. However, researchers started to realize that there are other TMDCs (e.g., MoSe_2_ and WS_2_) that offer better performances as compared to MoS_2_ (e.g., greater charge storage ability [45]). This further led to a gradual increase in research articles that studied the use of other potential TMDCs, since the year 2018. Figure 2 outlines the schematic diagrams of each gas adsorption process. In general, the mechanism of these adsorption processes is mainly driven by the charge transfer process between the adsorbates and the TMDCs-based adsorbent, where the charge movement is dependent on the nature of the gas adsorbate (i.e., oxidizing or reducing) [46]. For instance, carbon monoxide (CO) in Figure 2a is an example of reducing gas. Due to the existence of lone pair on the carbon atom, CO will donate electrons to the TMDC surface, which further cause the CO to be chemisorbed on the surface. Whereas, oxidizing gas such as nitrogen dioxide (NO_2_) in Figure 2c, will uptake the electron from the surface instead (mainly due to the existence of unpaired electron on the nitrogen atom). To note, such electron movement will lead to the deviation in electrical conductivity of the TMDC materials [47]. The following subsections discuss various gas adsorption processes.

### 2.1. Adsorption of Carbon Monoxide (CO)

Carbon monoxide (CO) is by far the most hazardous greenhouse gases, which is 210 times easier to bind with hemoglobin as compared to the oxygen [49]. To date, numerous studies have discovered the use of TMDCs as a catalyst to adsorb CO and further convert it into other products via catalytic reaction. For instance, Li et al. [50] proposed the use of aluminum oxide-doped MoS_2_ as the nanocatalysts to promote the CO methanation process (CO + 3H_2_ → CH_4_ + H_2_O) and enhance the stability of this catalytic reaction (MoS_2_ with 25.6% of Al_2_O_3_ provides the greatest methanation stability). The schematic diagram of this process is shown in Figure 2a. The incorporation of metal-based promoters, in this case, Al_2_O_3_ powder has effectively reduced the aggregation effect between MoS_2_ (lower tendency to pore blockage) [50]. Aside from that, researchers also adopted the density functional theory (DFT) calculation to explore the theoretical potential of each TMDC as the alternative catalyst for CO oxidation (CO + O_2_ → CO_2_ + O) and CO dissociation. For instances, studies showed that metal-doped MoS_2_ (e.g., Au_29_, Cu, Ag, Co, Rh, Ni, Ir, and Fe [51,52,53]) can significantly enhance the O_2_ dissociation (which is the essential step for CO oxidation). On the other hand, the latter five mentioned metals possess the greatest potential as they favor CO dissociation [52]. More recently, application of other TMDCs (e.g., Pt and Au nanoclusters with WSe_2_, where the carbon atom in CO will be strongly adsorbed on the Pt/Au decorated WSe_2_ monolayer [54]; Rh-doped MoSe_2_ [55]), in CO adsorption have also been carried out. Since CO dissociation is the first step of the Fischer–Tropsch process [56], the aforementioned TMDC-based nanocatalysts can be served as the substituent cheap and stable catalyst to convert CO into other valuable liquid hydrocarbons.

### 2.2. Adsorption of Water Vapor (H_2_O)

Water vapor (H_2_O) is the largest contributor to the current global warming issue (i.e., roughly accounted for 60% of the entire warming effect [57]). Numerous works have proposed the use of TMDC-based nanocatalysts to convert H_2_O into clean hydrogen (H_2_) via hydrogen evolution reaction (HER) (see Figure 2b). In general, hydrogen atoms in H_2_O will be adsorbed on the active site of the HER nanocatalysts. With the aid of the doped metal, the dissociation of the O–H bond in H_2_O is enhanced. The gas is then desorbed as H_2_ [58,59]. This can mitigate the warming effect attributed to the water vapor content in the atmosphere, and at the same time, serves as an alternative greenway for hydrogen production. The past two decades ago, MoS_2_ was referred to as one of the most prominent alternatives to substitute the conventional HER catalysts [60]. Nevertheless, the commercialization of TMDC-based HER nanocatalysts is still hindered due to various technical challenges (e.g., poor HER stability under acidic conditions [61,62] and weak intrinsic conductivity [63]). Numerous works have discovered ways to enhance the HER performance of MoS_2_. Just to name a few, these works proposed to improve the HER activity by (i) coating the MoS_2_ with metals (e.g., palladium [64], aluminum [65], gold [66], platinum [67], etc.); (ii) using a novel electrochemical approach to deposit the MoS_2_ nanosheet [68]; (iii) introducing the use promising support for the MoS_2_ nanoparticles (e.g., multi-wall carbon nanotubes [69]); (iv) varying the choice of dispersion media used [70]; and (v) coupling with other electrocatalysts (e.g., molybdenum carbide [71]). On the other hand, some researchers also proposed the use of other TMDCs in HER applications. For instance, Wang et al. [72] proposed to use a low-cost MoSe_2_ nanosheet, which offers decent active sites for HER activity and better HER performance as compared to the aforementioned MoS_2_-based catalysts. On the other hand, Seok et al. [73] have reported the intrinsic activity for HER on MoTe_2_ nanosheet via the first-principle calculation and scanning tunneling microscopy (STM) study. Besides, numerous researchers also successfully developed promising WS_2_-based and WSe_2_-based electrocatalysts recently (e.g., Te–WS_2_ nanosheet by Pan et al. [74]; and Ni-WSe_2_ nanosheet by Kadam et al. [75]).

### 2.3. Adsorption of Carbon Dioxide (CO_2_)

Aside from water vapor, carbon dioxide (CO_2_) is another key contributor to climate change issues (accounting for more than 75% of the total emitted greenhouse gases [76]). To address this issue, some researchers have attempted to apply MoS_2_ nanosheets as an effective membrane [77] and adsorbent [78,79] to separate CO_2_ from the gaseous mixture. Sun et al. [78] reported that the adsorption force between CO_2_ and MoS_2_ varies according to the strength of the applied electric field. Generally, CO_2_ will adsorb on the surface when an electric field is applied and desorb when the electric field is relieved (see Figure 2d). This unique feature makes it become a potential carbon capture media. In addition to the carbon capture process, TMDCs can be modified so that the captured CO_2_ can be catalytically converted into other CO_2_-reduction products. Shi et al. [80] were the first to discover the potential of copper modified MoS_2_ nanosheets in the CO_2_ reduction process. The incorporation of Cu nanoparticles has enhanced the adsorption capability as CO_2_ can now be adsorbed not only on the MoS_2_ nanosheets but also on the doped metal nanoparticles [80] (see Figure 2d). This work found that most of the CO_2_ was converted into CO (i.e., faradaic efficiency (FE) = 33–41%), followed by methane (FE = 7–17%) and with a trace amount of ethylene (FE = 2–3.5%) [80]. More recently, a research project in Korea has discovered the potential of a composite catalyst that encompasses MoS_2_ nanosheets and *n*-type Bi_2_S_3_, in the CO_2_-to-CO photoreduction process [81]. DFT calculations of various modified-TMDC nanosheets (e.g., SnO_2_-loaded MoS_2_ nanosheet [82]; TiO_2_-doped MoS_2_ nanosheets [83]) for the CO_2_ reduction process were also conducted recently. Nevertheless, the research on the potential of other TMDCs in CO_2_ adsorption is still considered scarce. To date, only a few have discovered their potential for the CO_2_ sensor [84,85].

### 2.4. Sulphur Content Removal

Based on the air quality data observed by the NASA Ozone Monitoring Instrument (OMI) satellite, about 60% of the total sulphur emissions in 2018 were anthropogenic emissions [86]. Despite the recent advancement in renewable energy and air treatment technologies have reduced the global sulphur emissions (sulphur emissions in 2015 is about 30% less as compared to that of 1990 [87]), the health impact of sulphur content in the atmosphere can still be severe. Thus, the sulphur emissions (including SO_2_, H_2_S, etc.) must still be monitored and controlled closely. Wei et al. [56], based on the DFT simulation, have explored the adsorption performance of SO_2_ and H_2_S on the Ni-doped MoS_2_ monolayer material. To note, due to the large number of free electrons offered by the doped metals, the overall adsorption capability of the TMDC-based adsorbents are, therefore, gradually enhanced [88]. This study is then extended by other researchers, by testing the effect of other doped metal atoms (e.g., palladium [89], platinum [90], gold [89], and copper [91]). All these studies show that the metal-doped MoS_2_ monolayer materials can offer promising adsorption properties to SO_2_ and H_2_S gases, where both gases are chemisorbed on the surface with strong interactive forces. Similarly, various DFT studies have also been conducted to study the potential of using metal-doped MoSe_2_ nanosheets (e.g., rhodium-doped MoSe_2_ monolayer material [55] and palladium-doped MoSe_2_ monolayer material [92]) and WSe_2_ nanosheets (e.g., N-doped TiO_2_/WSe_2_ nanocomposite [93] and silver-doped WSe_2_ monolayer material [94]) in sulphur content removal applications. Dan et al. [95], on the other hand, studied the photocatalytic H_2_S splitting (H_2_S → H_2_ + S) on the proposed novel composite catalysts, MnS/In_2_S_3_-MoS_2_ based on first-principles calculations. This is an attractive and cost-effective way to generate clean H_2_ that can substitute conventional coal-derived hydrogen production [96]. In other words, this also indirectly contributes to pollution mitigation. The schematic diagram for the aforementioned sulphur removal process is shown in Figure 2e, where the desorption process can be conducted by altering the operating conditions (e.g., temperature and pressure).

### 2.5. Nitrogen Oxide (NO_x_) Removal

Nitrogen oxide (NO*_x_*) has adverse effects on human health and ecosystems. To date, numerous works have revealed the potential of applying TMDCs for NO*_x_* removal purposes. Notably, a novel Cu_2_O-anchored MoS_2_ nanocomposite developed by Yuan et al. [97], can achieve up to 53% NO*_x_* removal. In another research conducted in China, a similar NO removal rate (about 51%) can also be obtained using the MoS_2_-g-C_3_N_4_ nanocomposite [98]. In the same year, Xiong et al. [99] managed to develop a novel (BiO)_2_CO_3_/MoS_2_ photocatalysts that could achieve 50% NO removal for five consecutive cycles. The study was extended by incorporating the use of carbon nanofibers into nanocomposites fabrication [100]. The developed Bi_2_O_2_CO_3_-MoS_2_-CNFs nanocomposites can attain almost 70% of NO removal rate. Under visible light irradiation, the oxidative power offered by the nanocomposites is sufficient to oxidize the NO*_x_* into NO^3−^ (see Figure 2c). Besides, first-principle studies were also conducted to determine the metal-doping effects on the adsorption capabilities on MoS_2_ (e.g., (vanadium, niobium, tantalum)-doped MoS_2_ [101]), MoSe_2_ (e.g., palladium-doped MoSe_2_ [92] and rhodium-doped MoSe_2_ [55]), and WSe_2_ [102] monolayer materials to NO*_x_* gases.

## 3. Gas Sensing for Pollution Reduction

Gas pollution is commonly caused by direct greenhouse gases (CO, CO_2_, N_2_O, CH_4_, Fluorinated Gas, etc.) [103], indirect greenhouse gases (NH_3_, NO*_x_*, H_2_S, etc.) [104], and other traces of toxic gases. These gas pollutants may cause climate change, ozone pollution, and even threaten food security [105] Further adverse effects that are caused by these problems include reduced photosynthesis performances [106], increased chances of respiratory problems [107], and acid rain [108].

Gas sensing can be deployed in multiple locations to act as an environmental monitoring system [109]. More recent applications of gas sensing products include gas monitoring using drones [110], wearable gas sensing devices [111], and the internet of things (IoT) multi-gas sensing modules [112]. For gas sensing TMDC-material in the device, the most common working principle is for adsorption of gas particles towards the TMDCs for a charge transfer, giving a change of resistivity in material, then desorption from the TMDCs [113]. In this field, researchers are constantly searching for TMDC materials that can provide a very low limit of detection (LOD), high sensitivity, short response and recovery time [114,115,116].

### 3.1. NO_x_ Detection

NO*_x_* detection by TMDCs has received much research interest due to its effectiveness and applicability. The measurement of NO*_x_* in environmental monitoring is a common requirement [117] while the design of a perfect NO*_x_* sensor poses some challenges. Traditional SnO_2_ sensors have problems with the dual response towards oxidizing gases and reducing gases, giving low sensitivity when performing measurements in a multi-gas environment [118]. The selectivity of the gas between NO_2_ and NO is also a crucial challenge for NO*_x_* sensors [119].

A popular TMDC material for NO_2_ detection is MoS_2_ [113,120,121]. Earlier works [121] used CVD-grown MoS_2_ to detect NO_2_ gas at room temperature with LOD down to parts per billion (ppb) ranges, however, the recovery time was long. Consequently, a higher temperature was used to accelerate the desorption kinetics of the NO_2_ gas molecule [113,120]. Other researchers also created nanocomposites of TMDCs with materials such as graphene aerogel [122] and reduced graphene oxide (rGO) [123] to improve sensitivity and lower LOD. Pham et al. [115] used a red-light illumination to provide photon energy, which matches the bandgap of the MoSe_2_ sensor. The technique resulted in low LOD (25 ppb) in room temperature conditions with high sensitivity. Moreover, Liu et al. [114] synthesized a flower-like porous SnS_2_ NSs with edge exposed MoS_2_ nanosphere, which resulted in a very fast response time of 2 seconds. Group-10 noble TMDCs such as PtSe_2_ has also received much research attention due to their widely tunable bandgap and excellent performance in gas sensing [124]. From a first-principle study, Sajjad et al. [125] discovered that monolayer PtSe_2_ exhibited lower adsorption energy than MoS_2_ and graphene. For this, Yim et al. [126] used thin films of PtSe_2_ stacked on Si to achieve NO_2_ detection down to 9 ppb. Other TMDC materials such as WS_2_ [122] and MoTe_2_ [127] also showed that they have excellent NO_2_ sensing properties, and more research work is to be expected for different transition metals. A collection of NO_2_ sensing TMDC materials is tabulated in Table 1.

Although NO and NO_2_ are sometimes measured as total NO*_x_*, many academic studies require differentiation between NO from NO_2_ [133,134]. A distinct property between the two gases is that NO is a colorless gas while NO_2_ is reddish-brown in color [135]. Due to its reduced electron valence in NO, first-principle studies demonstrated that the adsorption energy of NO to be slightly higher than that of NO_2_ [136]. Nevertheless, the binding of NO with MoS_2_ is still considered one of the strongest in many gas molecules including CO, CO_2_, NH_3_, CH_4_, H_2_O, N_2_, O_2_, and SO_2_ [136]. This computational finding was verified by a few experiments that successfully detected NO with MoS_2_ materials down to the ppm level [137,138,139]. Although there is not much research work that focuses on NO detection, it is demonstrated to give promising performances using TMDC material (see Table 2).

### 3.2. Ammonia Detection

Ammonia is an indirect greenhouse gas, as it is quickly soluble by water vapor or rain in the atmosphere and goes down to the land to be readily used as fertilizers by plants [141]. The lifespan of this alkaline and reactive gas in the atmosphere is short, around 1 day [142]. Nevertheless, direct exposure to ammonia gas from its emission source can still cause health issues to humans [143]. Gases such as NO_2_ are electron-accepting, while NH_3_ donates electrons to the TMDCs surface [113] to give a change of resistivity in the material (see Figure 3a,b). Due to their different adsorption mechanism, NO_2_ and NH_3_ are two flagship gases to study the generic effect of gas sensing [113,144,145].

Many works have also indicated that TMDC materials have excellent properties to act as excellent ammonia gas sensors [145]. For example, Cho et al. [146] used a 2D graphene/MoS_2_ heterogeneous structure to improve the sensitivity of the sensor towards gas molecules. Burman et al. [139] studied the effect of vacancy sites on MoS_2_ sensors and proposed a UV-treated sensor that can be synthesized from powder. Other TMDC materials such as MoTe_2_ [147] and WS_2_ [148] also showed performances of being able to detect ammonia gases down to ppm levels. Even artificial neural networks were used to post-process signals from TMDC ammonia gas sensors to improve gas concentration predictions [144]. A few of the significant works of applying TMDC materials as ammonia gas sensors are found in Table 3.

### 3.3. Volatile Organic Compound (VOC) Detection

Volatile organic compounds (VOC), commonly known as aromatic hydrocarbons fractions, are organic chemicals that have high vapor pressure in atmospheric conditions. VOC are studied to be emitted in landfills [153], newly renovated buildings [154], industrial refineries [155], and other leakage sources. Long-term exposure to VOC can lead to dysfunction of central nervous systems, memory loss, and cause congenital anomalies for reproduction [156]. VOC sensors are important for environmental detection [154], industrial emission control [157], and even for cancer diagnosis [158].

The challenge of designing sensors for VOC is that VOC is not a single gas molecule but can consist of multiple molecules (such as benzene, ethylene glycol, formaldehyde, etc.) with slightly different properties. Barzegar et al. [159] represented the VOC molecule behavior by using xylene isomers and methanol, showing high potential for adsorption between VOC molecules and a Ni-decorated MoS_2_ sensor. A thiolated ligand conjugate MoS_2_ sensor was also shown to exhibit excellent sensitivity down to a concentration of 1 ppm [160]. The work represented VOC by studying a combination of toluene, hexane, ethanol, propanal, and acetone gas molecules while demonstrating the conjugation of thiolated ligand improves the charge carrier density of the sensor (see Figure 4). A recent work from Tomer et al. [161] demonstrated that a cubic Ag(0)-MoS_2_ loaded g-CN 3D porous hybrid had multifunctional abilities towards sensing VOC by studying *n*-butanol, isopropanol, benzene, and xylene gas molecules. Short response time and recovery time of 7–15 s and 6–9.5 s respectively were obtained for all VOC molecules at 175 °C. Zhao et al. [162] presented an optical VOC microsensor using a photonic crystal cavity integrated with MoS_2_. VOC molecules of acetone, methanol, ester, dichloromethane, and methylbenzene were studied and the optical microsensor resulted in a LOD of 2.7 ppm, response time of 0.3 s, and recovery time of 100 s.

### 3.4. Detection of Sulphur Gases and Other Gases

The unique properties of TMDC materials also showed many preliminary potentials for the detection of sulphur gases and other emissions. Chen et al. [163] demonstrated from density functional theory (DFT) that MoS_2_ monolayers exhibit great adsorption energy towards SF_6_ decomposition (which includes SO_2_, SOF_2_, SO_2_F_2_, H_2_S, and HF). The adsorption of SF_6_ decomposition by monolayer PtSe_2_ was also studied by DFT methods and found to give excellent adsorption energies [164]. Park et al. [165] experimentally demonstrated that a Pt nanoparticle decorated MoS_2_ gas sensor can achieve high sensitivity detection of H_2_S down to 30 ppm. Recent work from Yang et al. [166] demonstrated that Ni- and Cu-embedded MoS_2_ monolayers can improve the adsorption energy of SO*_x_* and O_3_ molecules compared to that of the pristine MoS_2_.

TMDCs also demonstrate adsorption and sensing ability for CO gas. The work of Ma et al. [167] studied the potential of sensing CO and NO gas molecules by doping metal particles on MoS_2_ monolayer. The doping of Au, Pt, Pd, and Ni has shown to alter the transport property of MoS_2_, giving better adsorption energy for CO and NO gas detection. Recently, Yang et al. [91] presented a strategy of using Ti doping and the application of an electric field to improve adsorption energies in MoSe_2_ and MoS_2_ materials. The work concluded that Ti doping strategy improved the sensitivity of both materials towards CO and NO gas detection. Another recent work from Shen et al. [168] provided a study of using a borophene/MoS_2_ heterostructure to detect small gas molecules (CO, NO, NO_2_, and NH_3_), which exhibited different resistive properties towards these molecules when changing voltage direction (see Figure 5). Modulation of MoS_2_ by antisite doping and strain also showed improved gas detection sensitivity [169]. Ma et al. [170] studied the effects of defects on WSe_2_ monolayer sensor and found that Se vacancies can improve sensing ability of H_2_O and N_2_ molecules. Further interest gas detection also extends towards Cl_2_, PH_3_, AsH_3_, BBr_3_, and SF_4_ gas molecules on MoS_2_ [171]. More materials will be expected to be discovered for gas detection applications in the future.

## 4. TMDC Materials for Wastewater Treatment

The United Nations has highlighted that 80% of wastewater is released back to the environment without sufficient treatment [172]. The effect of industrial development and human activities has released many pollutants into the water. The major water pollution sources come from various source such as industry waste [173], sewage and wastewater [174], oil leakage [175], chemical fertilizer and pesticides [176], and mining activities [177]. These pollutants require high oxygen for oxidation decomposition, which reduces the dissolved oxygen level in the water that will damage the aquatic ecosystem [178]. Wang et al. [179] reviewed that many technologies including adsorption, ion exchange, membrane filtration, chemical precipitation, and electrochemistry have been widely used in water treatment. Despite the application of various technologies, quality water supply remains unsustainable [180]. Many researchers have reported promising outcomes with the application of nanomaterial such as TMDC-based material in water treatment technology [181].

### 4.1. Adsorption for Wastewater Treatment

Among wastewater treatment technologies, the adsorption is considered the most inexpensive, fast and simple operation method [182]. Recently, researchers are focusing on the application of TMDC-based material in adsorption technology. That MoS_2_ nanosheet is used in the adsorption process mainly due to its large surface area, excellent chemical and thermal stability and environmentally friendly [183]. Many application of TMDC-based material has been successfully demonstrated with an effective outcome including the removal of methyl orange [184], Rhodamine B (RhB) [185], and Congo Red [186]. Li et al. [187] stated that the application of MoS_2_ adsorbent can increase the adsorption capacity in removing organic dye. An alternative facile oxidation strategy was proposed by Li et al. [182] to synthesize tungsten disulphide/tungsten trioxide (WS_2_/WO_3_) heterostructures to remove RhB molecules from wastewater (see Figure 6a). This strategy has contributed to higher adsorption capacity through manipulating the surface property. Massey et al. [188] further added that the hierarchical microsphere of MoS_2_ nanosheets has demonstrated high adsorption capacity with 297, 216, and 183 mg/g of methylene blue, rhodamine 6G, and fuchsin acid dye respectively as compare to conventional absorbent such as activated carbon. Most importantly, the hierarchical microsphere of MoS_2_ nanosheets can be regenerated effectively without affecting the performance of adsorption capacity.

Kumar et al. [189] concluded that the synthesis of MoS_2_ with magnetic nanoparticles can be effective adsorption material especially in removing heavy metal such as chromium. The efficiency of chromium removal is highly dependent on the pH of the solution and the adsorption of Cr(VII) and Cr(III) can be selected by changing the pH (see Figure 6b). Besides that, the regeneration of absorbent does not reduce the efficiency of chromium uptake significantly.

On top of that, the CeO_2_-MoS_2_ hybrid magnetic biochar (CMMB) also exhibits a strong magnetic ability to remove lead(II) (Pb(II)) and humate from water treatment [191]. The CMMB can remove >99% of Pb(II) and humate within 6 h. The application of MoS_2_/thiol-functionalized multiwalled carbon nanotube (SH-MWCNT) has also displayed high adsorption capacity for heavy metal removal [177]. However, the spent adsorbent can be further used for photocatalytic and electrochemical applications.

Furthermore, the synthesized MoS_2_-coated melamine-formaldehyde (MF@MoS_2_) sponges that exhibit a superhydrophobic and superhydrophilic characteristic (see Figure 6c) demonstrated a highly selective adsorption capacity [190]. The MF@MoS_2_ sponges have high absorption performance for oil and organic solvent and water-soluble dye. It also exhibits high discoloration efficiency of 98% methyl orange within 10 min. Wan et al. [192] added that the MF@MoS_2_ can be modified from room temperature vulcanized silicon rubber.

### 4.2. Membrane Technology in Wastewater Treatment

In water treatment systems, the typical microporous membrane pore size is about 0.1–5 µm, which limits membrane technology on water purification [179]. The development of nanoporous membranes exhibited high filtering performance in dealing with most of the pollutants inducing microbes, organic molecules, heavy metal, and salts.

Heiranian et al. [193] highlighted that the single-layer MoS_2_ membrane with a full Mo pore (see Figure 7a) exhibited 88% of ion rejection. The work also demonstrated that the water flux was two to five orders of magnitude greater than other known nanoporous membranes. This technology is crucial to replacing the reverse osmosis (RO) membrane, especially in water desalination processes. Later, Kou et al. [194] discussed that a 0.74 nm nanopores in monolayer MoS_2_ membranes should be compatible with Debye screening length for the electrostatic interaction and lesser than the mean free path of molecules in water. Recent work from Kozubek et al. [195] proposed using irradiation with highly charged ions (HCIs) to create pores in MoS_2_. They found that pore creation efficiency has a linear relationship with potential energies for pore radius of 0.55–2.65 nm. Kozubek et al. [195] added that the HCI method has parallel writing capabilities, giving a high potential for mass production.

In the application of water desalination, Ma et al. [196] stated that the synthesis of MoS_2_/GO membrane resulted in an enhanced water flux (from 8.83 to 48.27 L·m^−2^∙h^−1^) with improved salts removal capacity (from 54.32% to 96.85%). The MoS_2_/GO membrane also exhibited different mechanism for dyes and ions (see Figure 7b) where dyes are physically sieved while ions have a Donnan effect due to existing trapped ions within interlayer spacing [196]. Gao et al. [197] demonstrated the performance of MoS_2_ membrane’s performance in rejecting 100% of methylene blue dye. The work highlighted that the increase in nanosheets to the membrane will reduce permeability and increase dye rejection rate.

### 4.3. Photocatalyst Technology in Wastewater Treatment

As the majority of water treatment facilities are located outdoors, the availability of solar energy can be useful for photocatalytic technology in water treatment. Chu et al. [198] reported that the development of hexagonal 2H-MoSe_2_ photocatalyst exhibits outstanding photo-absorption and photocatalytic reaction in reducing hexavalent chromium (Cr(VI)) under ultraviolet (UV) light. The 2H-MoSe_2_ yields 99% Cr(VI) reduction rate with the presence of UV light. Mittal and Khanuja [199] revealed that MoSe_2_ is a good photocatalyst not only for Cr(VI) but also for methylene blue (MB) and RhB. The MoSe_2_/strontium titanate (SrTiO_3_) heterostructure also showed great potential as a wastewater treatment photocatalyst with a degradation rate of methyl orange at 99.46% under the optimum loading weight of 0.1 wt.% under UV light [200]. It is also found that there is no significant loss in the performance of MoSe_2_/SrTiO_3_ after reuse for 6 times. The working principle of MoSe_2_/SrTiO_3_ photocatalyst is demonstrated in Figure 8.

## 5. Fuel Cleaning

Fuels are means of transfer for energy in our ecosystems. Most fuels have many imperfections related to conversion inefficiencies [201] and fuel impurity (including sulphur, nitrogen, aromatic, etc.) [202]. Direct usage of crude fuels with such impurity content will lead to an increase in gas emissions [203,204]. Most refineries for oil-based fuel carry out fuel cleaning processes such as desulphurization, dearomatization, denitrogenation, and deoxygenation [202,205,206] to remove such impurities before the fuel is being converted. The use of impurity removal technology generally depends on the techno-economics and fuel costs during application [207]). Fuels with a high impurity such as sulphur content can potentially cause an elevated particulate emission [208] while causing catalyst poisoning in subsequent systems [209].

### 5.1. Fuel Hydrodesulfurization

Hydrodesulfurization is the process of reduction of sulphur content from fuel such as diesel [202] by catalytically reacting them with hydrogen. Paul et al. [210] studied the mechanism of vacancy formation on the MoS_2_ catalyst in the hydrodesulfurization process, demonstrating the activation energy of 0.5 eV. TMDC materials such as Co/Ni promoted MoS_2_ were popular for the use of hydrodesulfurization due to its high activity, good selectivity, resistance to deactivation, and regeneration ability [211]. Commonly, TMDC-based catalysts are used for hydrodesulfurization in the oil and gas industry for high-efficiency sulphur removal from naphtha [212]. For the hydrodesulfurization of naphtha, a recent work from Mahmoudabadi et al. [213] achieved 100% conversion efficiency using MoS_2_ quantum dots nanocatalyst under the pressure of 15 bars, temperature of 280 °C and liquid hourly space velocity (LHSV) of 4 h^−1^. The direct use of unsupported Ni/MoS_2_ catalyst was reported [214] to remove furfurylamine (FA) and dibenzothiophene (DBT). Liu et al. [215] demonstrated that for a MoS_2_/NiMo catalyst, the higher the number of MoS_2_ slabs being stacked, the better the hydrodesulfurization selectivity for DBT. Additionally, Rangarajan et al. [216] studied the preferred active sites of a metal-promoted MoS_2_ and found that organosulfur and organonitrogen compounds bind weaker on sites with exposed metal on the corner and the sulphur edge of MoS_2_. Hydrodesulfurization of DBT by various morphologies of MoS_2_ catalyst was also studied by Tye and Smith [217]. This work showed that exfoliated MoS_2_ has the highest conversion, compared to the commercially crystalline MoS_2_ powder and MoS_2_ derived from soluble Mo precursors. While selectivity was correlated to the edge sites of MoS_2_. A 3D NiS-MoS_2_/Graphene nanohybrid was also shown to give a high DBT conversion rate of 82.6%, showing potential for 3D composites. Recent work from Abbasi et al. [218] demonstrated that a cobalt-promoted MoS_2_ achieved more than 80% conversion for the application of diesel, showing potential for the transfer of high-performance TMDC catalysts from oil and gas to diesel/biodiesel industries.

Besides hydrodesulfurization to remove DBT, the removal of thiophene was also demonstrated to be feasible with MoS_2_ catalysis by DFT methods [219]. Kaluza et al. [220] synthesized a MoS_2_ catalyst supported on mesoporous alumina that gave a thiophene conversion of more than 60% at around 400 °C. For the removal of carbonyl sulphide (COS), Liu et al. [221] used a microwave activated MoS_2_/graphene catalyst that achieved over 90% conversion for temperatures over 280 °C. Later, the use of a monolayer MoS_2_ anchored on reduced graphene oxides [222] has shown to give over 90% conversion and a reduced temperature at about 180 °C.

### 5.2. Fuel Hydrodeoxygenation

Hydrodeoxygenation is the removal of oxygenated components from the fuel by reacting it with hydrogen to form water [206]. Liu et al. [223] found that a MoS_2_ monolayer doped with an isolated Co atom can effectively reduce hydrodeoxygenation of 4-methylphenol to toluene from 300 °C to 180 °C while maintaining high conversion and selectivity of 97.6% and 98.4% respectively. From a DFT perspective, Li et al. [224] demonstrate a generic structure of Co atom vacancy in MoS_2_, claiming that the catalyst can activate hydrodeoxygenation reactions at lowered temperatures. Recent work from Wu et al. [225] adsorbed Co oxide at the edge of MoS_2_ with sulphur defects and formed a Co-MoS_2−*x*_ catalyst that converts lignin-derived phenols (4-methylphenol) to arenes with 97.4% conversion and 99.6% toluene selectivity at a mild temperature of 120 °C (see Figure 9a). For direct industrial application, palm kernel oil was converted with a yield over 90% to jet fuel-like hydrocarbon (see Figure 9b) via a Ni-MoS_2_/*γ*-Al_2_O_3_ hydrodeoxygenation catalyst [206]. The work showed high potential for converting biomass-based fuel to high-value fuels using TMDC-based catalysts. Moreover, Co- and Ni- promoted MoS_2_ catalyst was also used to co-process diesel and vegetable oil via hydrodeoxygenation reaction [226]. Alvarez-Galvan et al. [227] demonstrated that transition metals phosphides were useful as a hydrodeoxygenation catalyst for waste cooking oil to green biodiesel. More research work is required to explore the possibility of TMDC materials in this field.

For catalytic deoxygenation of alkali lignin into bio-oil, Li et al. [228] proposed the use of a flower-like hierarchical MoS_2_-based composite catalyst that achieved a lignin conversion of 91.26% and bio-oil yield of 86.24%. Later, work from Zhou et al. [229] synthesized a series of MoSe_2_ catalysts for the conversion of alkali lignin into bio-oil. The work resulted in 96.46% conversion of lignin and 93.68% yield of bio-oil. A recent review by Porsin et al. [230] revealed that sulphide catalysts such as MoS_2_ can also convert fatty acid triglycerides to motor oil by hydrodeoxygenation. As for wider fuel cleaning applications, TMDC materials have also been reported to purify gas fuels such as hydrogen and methane [231] showing high potential in diverse fuel applications in the future.

## 6. CO_2_ Valorization and Conversion

Valorization of carbon dioxide (CO_2_) to raw chemical materials and clean fuels is an opportunity for the artificial carbon cycle, which contributes to the mitigation of global warming and alleviates the usage of fossil [232]. Over the decades, enormous efforts in searching of alternative technologies to mitigate the CO_2_ emissions through CO_2_ capture from concentrated industrial exhausts [233,234], sequestration of CO_2_ in the underground [235,236] and conversion of CO_2_ to energy-rich fuels powered by renewable energy resources [237] have been discovered. Throughout all these methods, CO_2_ molecules are not solely can be removed from the atmosphere but also can be converted into value-added chemicals such as methanol, formic acid, methane, and syngas [238,239,240]. Recently, electrochemical conversion of CO_2_ to value-added chemicals through the principal of CO_2_ electrochemical reduction reaction (CO_2_ERR) as shown in Figure 10a [241], has emerged as a comparative alternative to its counterparts such as biochemical and thermochemical technologies [242,243,244]. Alternatively, semiconductor-based photocatalysis of CO_2_ reduction has been another frontier in CO_2_ conversion [245]. When a light source with appropriate photoenergy is illuminated on this photocatalyst, electron-hole pairs are formed [246,247]. With the migration of the electron-hole pair to the surface, the release of energy from within can reduce surface adsorbed CO_2_ [246]. Another interesting type of photocatalyst is the Z-scheme photocatalyst, which performs charge transfer between a reduction semiconductor catalyst and an oxidation semiconductor catalyst, potentially giving higher performances, efficiency, and synthesis possibilities [248]. The general working principle of photocatalyst for CO_2_ conversion is illustrated in Figure 10b.

### 6.1. Conversion of CO_2_ to Syngas and Other Gases

To achieve this attractive blueprint, the key issue is to develop electrocatalysts with high activity, superior selectivity, highly durable, environmentally friendly as well as low cost [250,251,252]. In 2014, the first pioneering work of elucidating the TMDCs materials as advanced electrocatalysts for CO_2_ reduction was by Asadi and coauthors [253]. They have claimed that MoS_2_ material serves as a highly efficient electrocatalyst for the conversion of CO_2_ to syngas. The combination of the edge states of MoS_2_ in contact with ionic liquid solvent electrolytes has served as a new paradigm for CO_2_ reduction, providing the advantage of favorable electronic properties of MoS_2_ and an electrolyte that transfer the CO_2_ molecules to the active sites for reaction. This study is a breakthrough in this field, in which the performance of the catalytic activity for CO_2_ reduction reaction is far exceeding than other conventional catalysts such as carbon nanotubes, graphene, noble metal carbides, and transition metals.

Recent research work of TMDCs in converting CO_2_ to interesting gas products using photo- and electro-catalysis methods are shown in Table 4. Up to date, most reports of TMDCs in this field are focusing on the conversion of CO_2_ to CO, suggesting that there is still a huge potential of different electrochemical reaction pathways using TDMCs that can be explored in the near future such as the conversion of CO_2_ to CO and H_2_ to CH_4_. Wang et al. [254] synthesized a marigold-like SiC@MoS_2_ nanoflower for the conversion of CO_2_ and H_2_O to CH_4_ and O_2_ using no sacrificial agents while operating within the visible light spectrum (see Figure 11a). The work reported production rates of 323 and 621 µL∙g^–1^∙h^–1^ for CH_4_ and O_2_ while maintaining stable characteristics for 40 h. Asadi et al. [255] also have synthesized a series of TMDCs (e.g. WSe_2_, WS_2_, MoSe_2_, and MoS_2_) using the chemical vapor transport (CVT) growth technique followed by liquid exfoliation for electrochemical CO_2_ conversion using ionic liquid electrolyte (EMIM-BF4). By benchmarking with the bulk Ag and Ag NPs, the current density of TMDCs are more than tenfold (130–330 mA cm^−2^) of the current density of Ag (3.3 mA cm^−2^) and Ag NPs (10 mA cm^−2^) with 90% CO. Overall, WSe_2_ NFs gave the best performance and exhibited a current density of 18.95 mA cm^−2^ with a high turn-over frequency of CO of 0.28 s^−1^ at an overpotential of 54 mV. This can be related to the intrinsic properties of WSe_2_ (very low work function and high-volume surface area).

In addition, a lab-scale synthetic custom-builds wireless setup for photochemical studies of WSe_2_/IL is also being developed by this group [255]. The build-up of the artificial leaf that performed the photosynthesis process is investigated to further confirm the CO_2_ conversion rate efficiency (see Figure 11b). The system comprises of three main components: (i) harvest light using two amorphous silicon triple-junction photovoltaic cells in series, (ii) CO_2_ reduction using WSe_2_/IL on the cathode and lastly, and (iii) evolution of oxygen using cobalt (CoII) oxide/hydroxide in potassium phosphate electrolyte [265,266]. Surprisingly, the cell managed to function continuously for more than 4 h before corrosion happens at the transparent indium tin oxide layer on the anode, proving that this theoretical experiment is workable and insightful, which could be extended to the technology development scale in near future. Further studies on the defect engineering for CO_2_ reduction catalysts were also discussed by Wang et al. [267] to improve functionality by defect engineering (such as holes, doping, vacancies, phase change, edge, grain boundary, lattices distortion, and other methods). For this, MoS_2_ with substitutional defects on the edge was studied [253] and showed to give better current density than Au, Ag, Cu nanopowder, and the polycrystalline catalyst. Ji et al. [268] suggested that group 10 TMDCs have a high potential for CO_2_ reduction as they do not suffer from OH poisoning. The work proposed that the reaction energies for CO_2_ reduction could be tuned by vacancy densities.

### 6.2. Direct Conversion of CO_2_ to High-Value Chemicals

Direct conversion of CO_2_ to high-value chemicals (such as intermediate chemicals, fuels, and alcohols) has a high potential for commercial value [269,270]. TMDCs have demonstrated properties for catalyzing CO_2_ into important alcohol, especially methanol [85,271]. Bolivar et al. [272] studied the use of a Pt/MoO_x_/MoSe_2_ electrocatalyst and proposed that oxygen-containing surface species lowers the electric potential for methanol production. Moreover, Tu et al. [273] have demonstrated the methanol production of TiO_2_ and 0.5 wt.% MoS_2_ nanocomposites was higher than Pt/TiO_2_, Au/TiO_2_, and Ag/TiO_2_ for the photocatalytic reduction of CO_2_. For this TiO_2_/MoS_2_ composites, Xu et al. [274] showed that electrons are transferred from MoS_2_ to TiO_2_ upon contact, therefore promoting the separation of charge carriers upon photoexcitation. In 2018, Francis et al. [275] have reported on the activity of single crystals and thin films of MoS_2_ for the reduction of CO_2_ dissolved in an aqueous electrolyte to yield 1-propanol, ethylene glycol, and t-butanol and hydrogen. At an applied potential of −0.59 V, the Faradaic efficiencies for the reduction of CO_2_ to 1-propanol are *ca.* 3.5% for MoS_2_ single crystals and 1% for thin films with low edge-site densities. WS_2_ quantum dots doped Bi_2_S_3_ nanotubes also demonstrated high (38.2 μmol∙g^−1^∙h^−1^) production of methanol while co-producing ethanol [85]. Many researchers have also performed the conversion of CO_2_ to high-value products such as methanol, acetaldehyde [249] and ethanol [276] by using TMDC catalysts (see Table 5).

## 7. Challenges and Future Prospects

The application of TMDCs in pollution reduction is a breakthrough as has high availability [30,31,32], high economic potential [280,281,282], and excellent efficiency [254,276] for the application of pollution reduction. For TMDCs availability and efficiency, we have already discussed them in Section 1 and Section 2, Section 3, Section 4, Section 5 and Section 6 (depending on application) respectively. In terms of commercial techno-economic analysis, a few works have also pointed out the advantage of using TMDC catalysts. Valle et al. [280] studied the economics of converting biomass-derived syngas to ethanol using an indirect circulating fluidized bed gasification (iCFBG) system via the KCoMoS_2_ catalyst. The work highlighted that the iCFBG system could reach a minimum ethanol selling price (MESP) of 0.74 $/L (USD) while technologies such as entrained-flow gasification have that at 0.98$/L. This MoS_2_-based catalyst was also found to be more commercially viable than the Rh-Mn/SiO_2_ catalyst due to their cheaper economical costs, although the latter has better energy efficiency [281]. Phillips [282] also concluded that the use of Co/MoS_2_ has potential to lower down ethanol MESP down to 0.267 $/L (USD) highlighting that Ru or Rh-based catalysts are economically expensive even at low concentrations. Nevertheless, authors acknowledge that the techno-economic analysis of cutting-edge TMDCs technologies is still generally unknown. Thus, more TMDCs research studies should include consideration of economic analysis. Furthermore, despite the unique physiochemical properties offered by the TMDCs (e.g., the large surface area, which acts as a two-edged sword [283], tunable absorption heterostructure [200,284], flexible, diversiform, and functional properties [146]), more exploration on the operating conditions, kinetic, and thermodynamic parameters should be carried out to further confirm its ability [285]. This is because most of the reviewed works are conducted under a controlled environment with a specially designed flow. Under this context, whether obtained simulation and experimental results reflect the actual situation remains uncertain. The main limitations of commercializing of TMDCs-based technology are:(i)Most research works are focusing on the development of the materials as an effective pollutant degradation catalyst during oxidation, neglecting that in reality, the number of organic pollutants is in a mixture that may affect the performance of the engineered material [286,287].(ii)The rate of product formation for TMDC-based technology, such as syngas and methanol from the conversion of CO_2_ is still far from the industrial scale, which is not sufficient to accommodate the global demand [286,287]. More research effort is required to improve efficiencies.(iii)High preparation/fabrication cost and poor uniformity of the materials in large-scale production [288]. More effort is required in sustainable synthesis and fabrication of TMDC-based technologies.(iv)The larger surface area offered by the TMDCs can enhance the adsorption capability, but also cause higher effects of environmental disturbance [283]. Precise defect engineering to improve material performances can also be carried out [225,267]. Optimization and data analytics on the design dilemma and defect engineering should be highlighted and properly studied.

As a future prospect, commercial validation on the performance on a larger scale needs to be carried out. The results obtained from the experiment can be validated by conducting techno-economic analysis, such as Monte Carlo analysis for economic and risk feasibility [289,290]. Furthermore, advanced machine learning methods can also be used to accelerate experimental designs [291,292], obtain optimal performances [293,294], and for the discovery of new TMDCs composite structures [295,296].

## 8. Conclusions

The unique and exclusive features of TMDCs (e.g., layered structure, tunable bandgap, unique optical, thermal and electrical properties, etc.) have been the main driving force that drives the researchers’ attention on exploring the potential of the 2D materials in pollution reduction applications. This work summarized the state-of-the-art applications of various TMDCs under the context of pollution mitigation (including (i) gas adsorption and removal, (ii) gas sensing, (iii) wastewater treatment, (iv) flue cleaning, and (v) CO_2_ valorization and conversion). In addition to the up-to-date progress of TMDCs research, this article also discussed some of the key challenges for the future commercialization of TMDC materials. Apparently, many of the reviewed research works have authenticated their substantial potential to substitute the existing pollution mitigation media. Nevertheless, the current applications are still restricted to a lab basis, where the deviation of the actual performance of TMDCs under larger-scale production remains as the research gap. To usher TMDCs into the next level of utilization (i.e., from the lab scale to the industrial scale), the following three research directions should be followed up, (i) techno-economic analysis (TEA) study, (ii) experimentation under more rigorous and realistic conditions, and (iii) experimental optimization for application purpose. The authors sincerely hope this review can serve as an insightful guideline that inspires more researchers to venture into this new and exciting cutting-edge field.

## Figures and Tables

**Figure 1 nanomaterials-10-01012-f001:**
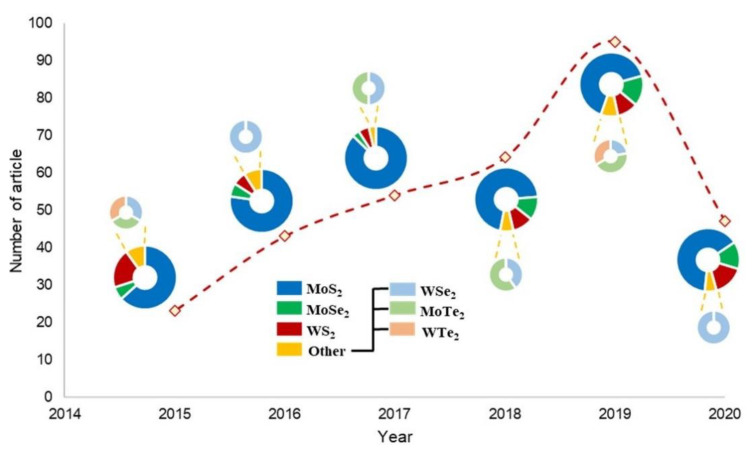
The number of research articles that applied transition metal dichalcogenides (TMDCs) in gas adsorption application between the year 2015 to April 2020, where pie charts represent the utilization percentage of various types of TMDC. Queried from Scopus Database [48]. Copyright Elsevier B.V., 2020.

**Figure 2 nanomaterials-10-01012-f002:**
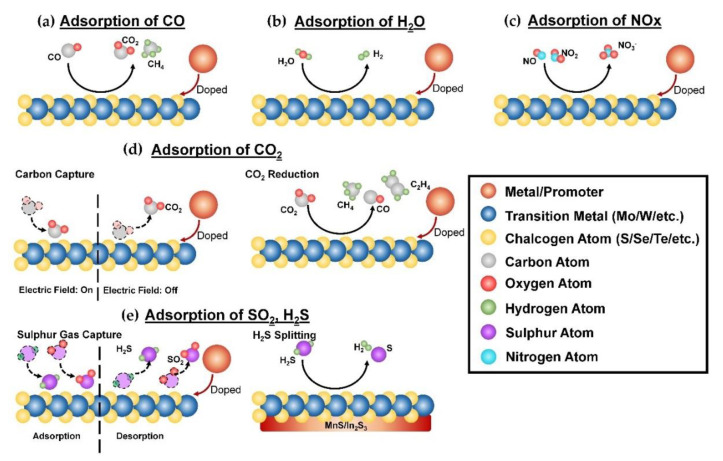
Schematic diagrams for the adsorption processes of (**a**) CO, (**b**) H_2_O, (**c**) CO_2_, (**d**) sulphur content, and (**e**) NO*_x_* with the use of TMDCs.

**Figure 3 nanomaterials-10-01012-f003:**
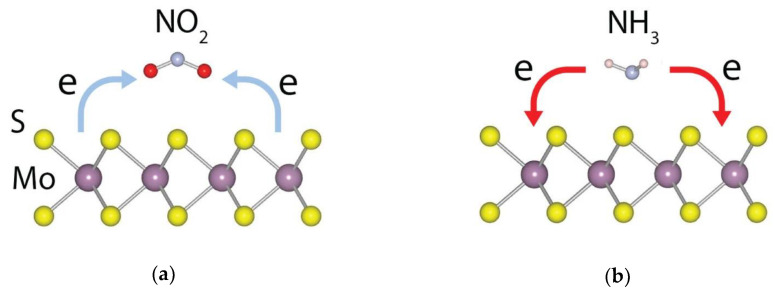
(**a**) Mechanism of electron-accepting NO_2_ gas molecule on MoS_2_ surface and (**b**) mechanism of electron-donating NH_3_ gas molecule on MoS_2_ surface (Reproduced with permission from [113]. Copyright 2015 Springer Nature Limited CC BY-NC-ND 4.0).

**Figure 4 nanomaterials-10-01012-f004:**
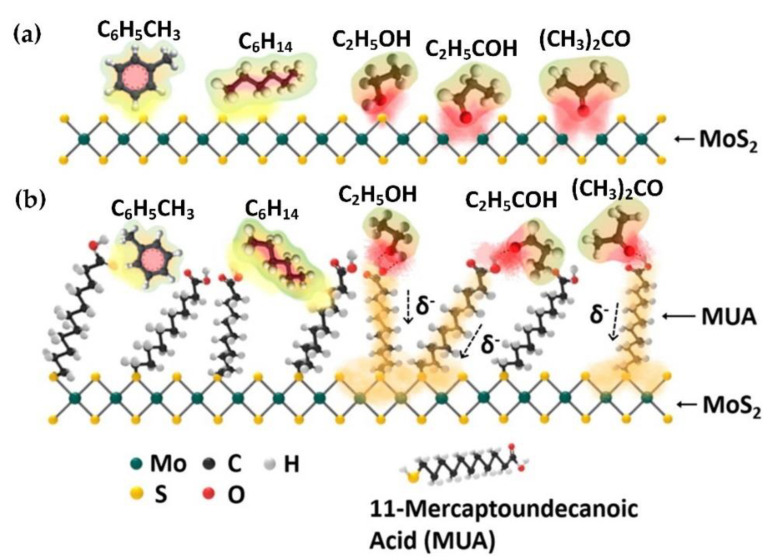
(**a**) Gas sensing mechanism for volatile organic compounds (VOC); toluene, hexane, ethanol, propanal, and acetone) gas molecule on primitive MoS_2_ surface and (**b**) improved VOC sensing performance by Mercaptoundecanoic acid (MUA)-conjugated MoS_2_ due to increased charge carrier density (Reproduced with permission from [160]. Copyright 2014 American Chemical Society).

**Figure 5 nanomaterials-10-01012-f005:**
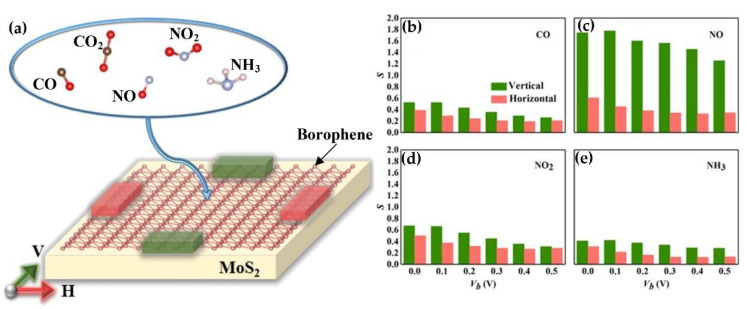
(**a**) Direction-sensitive borophene/MoS_2_ heterostructure for multiple climate gas detection. Sensitivity and directional bias voltage of (**b**) CO, (**c**) NO, (**d**) NO_2_, and (**e**) NH_3_ in the horizontal and vertical direction (Reproduced with permission from [168]. Copyright 2020 Elsevier B.V.).

**Figure 6 nanomaterials-10-01012-f006:**
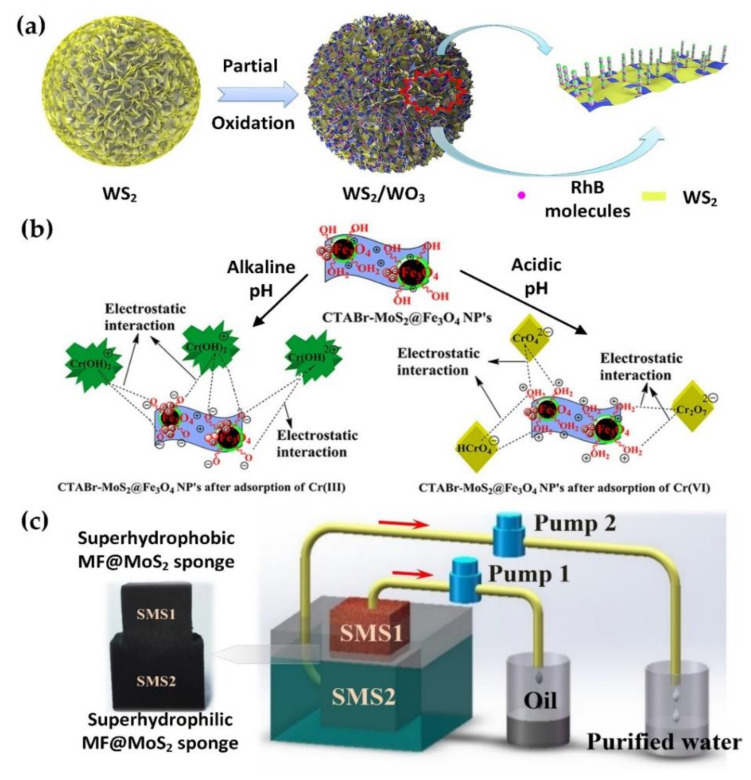
(**a**) Facile oxidation strategy to prepare a flower-like WS_2_/WO_3_ heterostructure for adsorbing RhB from wastewater (Reproduced with permission from [182]. Copyright 2020 MDPI AG CC-BY 4.0). (**b**) Bifunctional cetyltrimethylammonium bromide (CTABr)-coated MoS_2_-decorated Fe_3_O_4_ nanoparticles using pH to alter adsorption properties between Cr(VII) and Cr(III) in wastewater (Reproduced with permission from [189]. Copyright 2017 American Chemical Society CC-BY-NC-ND Under ACS AuthorChoice). (**c**) A superhydrophobic/superhydrophilic MF@MoS_2_ sponge for water/oil separation (Reproduced with permission from [190]. Copyright 2017 Elsevier B.V).

**Figure 7 nanomaterials-10-01012-f007:**
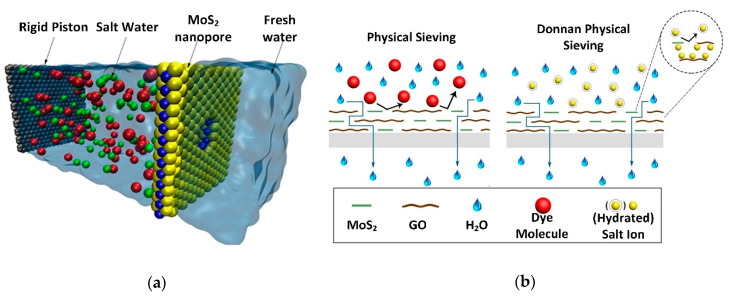
(**a**) Single-layer MoS_2_ water desalination membrane with Mo only pores (Reproduced with permission from [193]. Copyright 2015 Springer Nature Limited CC BY 4.0). (**b**) MoS_2_/GO nanosheets intercalated membrane for removal of dyes and salts (Reproduced with permission from [196]. Copyright 2020 Elsevier B.V.).

**Figure 8 nanomaterials-10-01012-f008:**
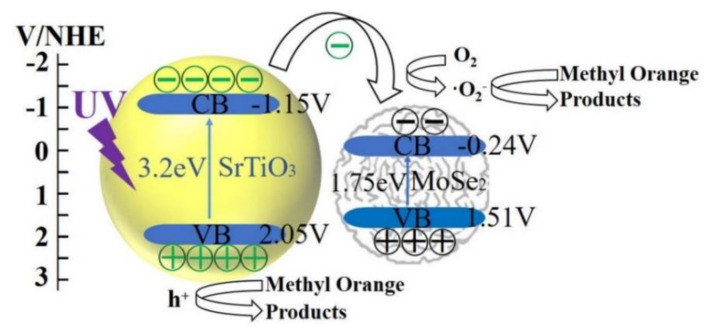
Mechanism of a MoSe_2_/SrTiO_3_ photocatalyst to oxidize and degrade methyl orange dyes in wastewater. UV irradiation causes electron transfer from the valence band (VB) to the conduction band (CB), which activates the separation process (Reproduced with permission from [200]. Copyright 2018 World Scientific Publishing Co Pte Ltd CC BY Under Open Access).

**Figure 9 nanomaterials-10-01012-f009:**
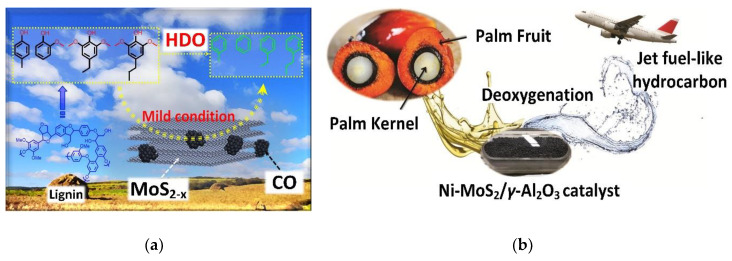
(**a**) Hydrodeoxygenation (HDO) of lignin-derived phenols into arenes using Co-MoS_2−*x*_ (Reproduced with permission from [225]. Copyright 2020 American Chemical Society) and (**b**) deoxygenation of palm kernel oil to jet fuel-like hydrocarbon using catalysts (Reproduced with permission [206]. Copyright 2017 Elsevier B.V.).

**Figure 10 nanomaterials-10-01012-f010:**
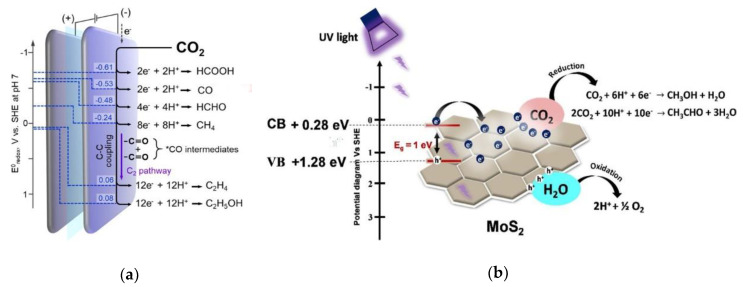
(**a**) Scheme of the electrochemical conversion of CO_2_ into value-added chemicals (Reproduced with permission from [241]. Copyright 2019 CC BY-NC 4.0 John Wiley and Son) and (**b**) general working principle of photocatalyst for CO_2_ conversion (Reproduced with permission from [249]. Copyright Elsevier 2019).

**Figure 11 nanomaterials-10-01012-f011:**
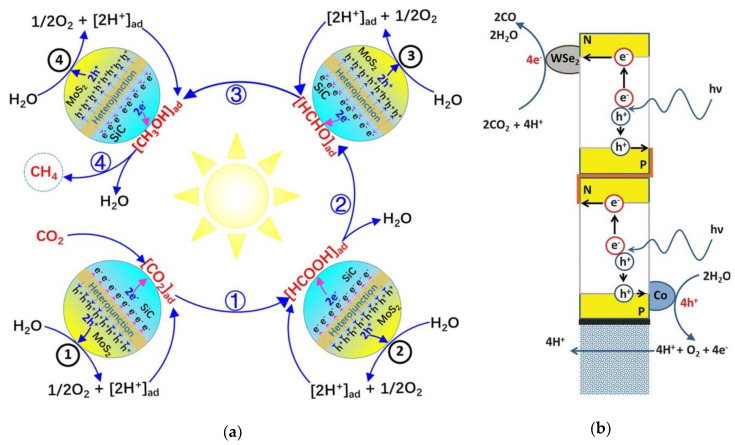
(**a**) Photochemical reaction pathway for the conversion of CO_2_ and H_2_O to CH_4_ and O_2_ under visible light using SiC@MoS_2_ nanoflower (Reproduced with permission from [254]. Copyright 2018 American Chemical Society) and (**b**) schematic of the WSe_2_-based artificial leaf cell design for the conversion of CO_2_ and H_2_O to CO and H_2_ (Reproduced with permission from [255]. Copyright 2016 The American Association for the Advancement of Science).

**Table 1 nanomaterials-10-01012-t001:** NO_2_ sensing TMDC materials and their performances.

Material	Target Gas	LOD	Condition	Remarks
MoS_2_/graphene hybrid aerogel [120]	NO_2_	50 ppb	RT to 200 °C	Response and recovery time < 1 min.
WS_2_ with AgNW functionalization [128]	NO_2_	25 ppm	100 °C	667% response compared to pristine 4L WS_2_ sensor.
MoS_2_ [121]	NO_2_	20 ppb	RT	Sensitivity of 194%/ppm
MoS_2_ [129]	NO_2_	120 ppb	RT, 100 °C	Peak to valley sensitivity > 50%.
MoS_2_ [115]	NO_2_	25 ppb	RT (with red light)	Sensitivity of 3300%/ppm.
rGO/MoS_2_ nanocomposite [123]	NO_2_	5.7 ppb	60 °C	Over 55% sensing response for NO_2_ at 8 ppm
ZnO/MoS_2_ [130]	NO_2_	200 ppb	RT (with monochromatic light)	Sensitivity at 29.3%/ppm, response time of 4.3 min, recovery time of 1.2 min.
SnS_2_/MoS_2_ [114]	NO_2_	25.9 ppm	RT	Response time of 2 s and recovery time of 28.2 s.
MoSe_2_ [131]	NO_2_	300 ppm	RT	Response time of 20 min and recovery time of 30 min.
MoSe_2_ [116]	NO_2_	10 ppm	RT (with UV light)	Response time <200 s.
WS_2/_WO_3_ composite film [132]	NO_2_	100 ppb	150 °C	Response time of 70 s and recover time of 120 s.
WS_2_/graphene aerogel [122]	NO_2_	10-15 ppb	RT	Response time of 70 s, recovery time of 300 s. (2 ppm)
MoTe_2_ [127]	NO_2_	20 ppb	RT (with UV of 2.5 mW/cm^2^)	Response time of 120 s with response of 18%.
PtSe_2_ [126]	NO_2_	0.9 ppb	RT	Response time of 1 s and recovery time of 4 s

**Table 2 nanomaterials-10-01012-t002:** NO sensing TMDC materials and their performances.

Material	Target Gas	LOD	Condition	Remarks
MoS_2_ deposited onto Si/SiO_2_ [137]	NO	0.8 ppm	RT	80% decreased response at 2 ppm.
MoS_2_ [138]	NO	100 ppm	RT, 50 °C, 100 °C (with UV light)	Response and recovery time below 600 s. Response at 25.63%.
UV-ozone treated MoS_2_ [139]	NO	20 ppm	125 °C	Stability issue over 120 ppm. Poor recovery.
WO*_x_*/WSe_2_ hybrid [140]	NO	0.3 ppb	RT	Response time of 250 s, S/N ratio > 10, sensitivity of 520%/ppm

**Table 3 nanomaterials-10-01012-t003:** Ammonia gas sensing TMDC materials and their performances.

Material	Target Gas	LOD	Condition	Remarks
MoSe_2_ [149]	NH_3_	50 ppm	RT	Response time of 2.5 min, recovery time of 9 min.
Graphene/MoS_2_ [146]	NH_3_	5 ppm	150 °C	Response time < 10 min, Recovery time < 30 min.
WS_2_ [148]	NH_3_	50 ppm	RT	Response time of 200 s, recovery time of 232.3 s.
MoS_2_ [121]	NH_3_	1 ppm	RT	Response time 5–9 min, recovery time < 15 min.
UV-treated MoS_2_ [139]	NH_3_	100 ppm	RT	Response time of 7 min, Recovery time of 12 min.
MoTe_2_ [147]	NH_3_	2 ppm	RT	Over 95% recovery using gate biases of 0 V and 20 V. Response time 10 min, recovery time 20 min.
MoS_2_/Co_3_O_4_ [150]	NH_3_	0.1 ppm	RT	Response time is 98 s, recovery time is 100 s.
PMMA-MoS_2_ [151]	NH_3_	1 ppm	RT	Sensitivity of 54%, response time of 10 s, recovery time of 14 s.
MoS_2_/VS_2_ [152]	NH_3_	5 ppm	40 °C	Recovery and response time both < 5 min.

**Table 4 nanomaterials-10-01012-t004:** Application of TMDCs in CO_2_ conversion to syngas and other gases.

Mechanism	Catalyst/Material	Details	Condition	Product
Electro-catalyst	2D nanoflake WSe_2_ [255]	50 vol% EMIM-BF4 in water	−0.764 V	CO
Electro-catalyst	Hierarchical MoS_x_Se_(2 − x)_ hybrid nanostructures [256]	0.1 M H_2_SO_4_ solution	−0.70V	H_2_
Electro-catalyst	^1^ Monolayer MoS_2_ [257]	-	-	CO
Electro-catalyst	5% niobium (Nb)-doped vertically aligned MoS_2_ [258]	50 vol% EMIM-BF_4_ in water	−0.8 V	CO
Electro-catalyst	molybdenum disulfide nanoflakes (MoS_2_ NFs) [259]	2.0 M C_5_H_14_ClNO	−2.0 V	CO
Electro-catalyst	Ultrathin MoTe_2_ [260]	0.1 M KHCO_3_	–1.0 V	CH_4_
Photo-catalyst	MoS_2_ nanoplatelet supported on few layer graphene [261]	Up to 60% conversion, 90% CH_4_ selectivity	React with H_2_, 250–500 °C	CH_4_, CO
Photo-catalyst	Marigold-like Si@MoS_2_[254]	Produced 323 μL∙g^−1^∙h^−1^ CH_4_, 23 μL∙g^−1^∙h^−1^ O_2_	React with H_2_O	CH_4_, O_2_
Photo-catalyst	Mesoporous TiO_2_ on 3D Graphene with Few-layered MoS_2_[262]	CO selectivity of 97% and yield of 93.22 µmol/g h	-	CO
Photo-catalyst	Z-scheme MoS_2_/g-C_3_N_4_ heterojunction [263]	Produced 58.59 μmol∙g^−1^ in 7 h.	25 °C, 100 kPa	CO
Photo-catalyst	MoS_2_/TiO_2_ heterojunction [264]	Produced 268.97 μmol∙g^−1^ CO and 49.93 μmol∙g^−1^ CH_4_	25 °C, 100 kPa	CO, CH_4_

^1^ Computational simulation using density functional theory (DFT).

**Table 5 nanomaterials-10-01012-t005:** Application of TMDCs in converting CO_2_ to high-value chemicals.

Mechanism	Catalyst/Material	Details	Condition	Product
Photo-catalyst	Undecorated 2D-MoS_2_ [249]	Produced 27.4 μmol∙g^−1^∙h^−1^ methanol and 2.2 μmol∙g^−1^∙h^−1^ acetaldehyde	0.5 M NaHCO_3_	Methanol, acetaldehyde
Photo-catalyst	MoS_2_/Bi_2_WO_6_ nanocomposites [277]	Produced 36.7 μmol∙g^−1^ methanol and 36.6 μmol∙g^−1^ ethanol (in 4 h)	Deionized water	Methanol, ethanol
Photoelectro-catalyst	Co-doped MoS_2_ NPs [278]	Produced 35 mmol∙L^−1^ methanol (in 350 min)	−0.64 V	Methanol
Photo-catalyst	Nano Ag decorated MoS_2_ nanosheets [271]	Produced 365.08 μmol∙g^−1^∙h^−1^	20 mL isopropanol,	Methanol
Electro-catalysis	MoS_2_ electrodes [275]	0.1 M potassium phosphate buffer	−0.59 V	1-propanol
Photo-Catalyst	WSe_2_/Graphene/TiO_2_ nanocomposite [279]	Produced 6.326 μmol∙g^−1^∙h^−1^ methanol	Distilled water with Na_2_SO_3_ reagent	Methanol
Photo-catalyst	MoS_2_/TiO_2_[273]	Produced 10.6 μmol∙g^−1^∙h^−1^ methanol	Distilled Water and methanol	Methanol
Photo-catalyst	WS_2_ quantum dots doped Bi_2_S_3_ nanotubes [276]	Produced 38.2 μmol∙g^−1^∙h^−1^ methanol and 27.8 μmol∙g^−1^∙h^−1^ ethanol	Ultrapure water	Methanol, ethanol

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
