# Peer review of "Transition Metal Dichalcogenides for the Application of Pollution Reduction: A Review"

_nanomaterials, 2020, doi:10.3390/nano10061012_

Round 1

Reviewer 1 Report

The review of  Transition Metal Dichalcogenides for the Application of Pollution Reduction is well organized and reflect current status of researches within the subject. 

In introduction, the authors tend to generalize. For example: "One typical feature of the electronic structures in most TMDCs is that the bandgap increases with the decreasing thickness"; "Lastly but importantly, TMDCs have abundant types and amount". These are partially true and valid for some material but not the whole TMDC class. References, justifications, and examples are needed.

In section 7, the authors stated " The application of TMDCs in pollution reduction is a breakthrough as it is more economic, higher availability and better efficiency as compared to other catalysts in pollution reduction." There is no evidence supporting that it is more economic, better efficiency, and especially they are not discussed in the text.

1 figure  every 4 pages: I would suggest to include more figures, if possible. 

Author Response

Thank you for your time and effort in reviewing our manuscript. We have made the corresponding revision to the manuscript as per your suggestion. The detailed "reply to reviewer" document is attached in the system.

Reviewer 2 Report

The main subject of the review is the application of transition metal-dichalcogenides (TMDCs) in dealing with environmental pollution. The review is predominantly focused on the potentials of TMDs for gas sensing, wastewater, and fuel cleaning. Due to the peculiar physical and chemical properties of TMDs as well as their well-established fabrication methods, they are of particular interest for many electronic, catalytic and energy conversion applications. The subject of this review can help to complement further technology roadmap of these materials for future environmental challenges. The manuscript is well written and provides an overview on the recent development of the topic. Before I can recommend this manuscript for publication, the following questions/comments should be addressed:

1-The authors mentioned the abundancy of TMDCs (few of them) in nature. The authors should also mention other preparation methods in particular chemical vapor deposition (CVD) based techniques (Adv. Phys. 1969, 18, 193). High-quality natural or synthetic bulk crystals can be used to fabricate few-layer TMDs via mechanical or liquid phase exfoliations (Chem. Soc. Rev. 2013, 42, 1934).

2-Recently, group-10 noble TMDCs such as PtSe2 have been reintroduced as new 2D materials, displaying many fascinating properties including widely tunable bandgap (Ang. Chem. Int. Ed. 2014, 53, 3015) and ultrahigh gas sensibility e.g. for detection of NH3, CO2, CO, NO, NO2 (Adv. Mater. Interfaces 2017, 4, 1600911; ACS Appl. Mater. Interfaces 2015, 7, 2189; ACS Nano 2016, 10, 9550). I would recommend including these reports in the discussion about gas sensing in section 2.

3-The authors addressed the modulation of bandgap in TMDCs. Among possible strategies for tuning bandgap, mechanical deformations via an applied strain or compression are of the most efficient techniques which should be also mentioned. The tensile strain causes significant changes in the electronic structure of TMDs and even cause the semiconductor-metal transition (Phys. Rev. B 2013, 87, 235434).

4-The authors addressed the application of TMD based membranes for wastewater treatment. This technology requires the fabrication of well-defined pores which is indeed a challenging task. It is, therefore, expected that authors provide some information on perforation of TMD materials (see e.g. J. Phys. Chem. Lett. 2019, 10, 5, 904; Phys. Chem. Chem. Phys. 2016, 18, 22210).

5-It is well known that defects engineering is one of the efficient methods to enhance the catalytic performance of TMDs. To be specific, recent reports demonstrated the role of defects in hydrogen evolution reaction (Nano Lett. 2016, 16, 2, 1097; Science 2007, 317, 100) and oxygen evolution reaction (ACS Catal. 2018, 8, 1683) of TMDs.

Minor comments:

6- In section 2.5, reference 25 was cited as Rhodium-doped MoS2 which is indeed Rhodium-doped MoSe2.

7- In section 3.4, it is written:” The doping of Au, Pt, Pd and Ni has shown to alter the transport property of MoS2, giving better adsorption energy for CO and NO gas detection.” The relevant citation is missing here.

Author Response

(The authors gave the same response as above.)

Reviewer 3 Report

Title:Transition Metal Dichalcogenides for the Application of Pollution Reduction: A Review

Authors: Xixia Zhang, Sin Yong Teng, Adrian Chun Minh Loy, Bing Shen How, Wei Dong Leong, Xutang Tao

Journal: Nanomaterials

Manuscript ID: nanomaterials-810168

The paper reviews the most recent advances in the field of transition metal dichalcogenides for environmental applications and, in particular, pollution reduction.

The paper is rather focused on the technological implications, and covers many aspects, from gas adsorption and removal, to gas sensing for pollution reduction, wastewater treatment, fuel cleaning, and finally CO2 valorisation and conversion. Challenges and future prospects are also outlined.

Although rather technical, the paper is easily readable, well written, clear. The figures are well conceived and well finished.

The topic is timely and of utmost interest, since it links the ongoing research on the widely explored field of two-dimensional materials and van der Waals heterostructures based on transition metal dichalcogenides to the fundamental technological challenge of reducing environmental pollution.

With these motivations, the paper undoubtedly warrants publication in Nanomaterials.

However, I suggest the authors to make some minor modifications, to improve it in some parts. Here I summarise my suggestions:

  • I really appreciate figure 2, because it clearly depicts the main chemical processes occurring when an external gas molecule adsorbs on the TMD surface. However, I cannot find a satisfactory discussion/comment of the processes drawn in the figure in the different subsections devoted each to a different molecule. I think that adding such a discussion would make the presentation even more valuable
  • related to the previous point, for the different molecules, I would add more details about the chemical/physical processes occurring at the TMD surface for the different molecules. For example, is there any physical/chemical property that is known or expected to change upon adsorption (e.g. surface work function) and which a future sensing device might be based on?
  • there are very few misprints (e.g. "Firstly, TMDCs have a high surface to volume ratio. IT OFFERS", "THESE gas pollutioN causeS"), it would be desirable a final, careful re-reading of the manuscript to correct them.

In conclusion, the paper offers a significant and clear contribution in the field of transition metal dichalcogenides applied to the fight to environmental pollution challenge. The clarity of the presentation offers to the readers of Nanomaterials an important and valuable perspective. My suggestion is for publication after the minor issues outlined above have been addressed.

Author Response

(The authors gave the same response as above.)

Reviewer 4 Report

This is a interesting review paper about the Transition Metal Dichalcogenides for the Application of Pollution Reduction. I have a following suggestions that authors may wish to revise the current content. First, the Optoelectronic properties of various transition metal dichalcogenides may be included with respect to their applications. Also, the more figures should be included for each section (eg. Sec. 3, 4 and 5) of the review.

Author Response

(The authors gave the same response as above.)

Round 2

Reviewer 2 Report

The authors have fully addressed all of my concerns in the previous report. Therefore, I can recommend the work for publication.